# Performance of Full-Component Coal Gasification Fine Slag: High-Value Utilization as Reinforcing Material in Styrene-Butadiene Rubber (ESBR) for Replacing Carbon Black

**DOI:** 10.3390/polym16040522

**Published:** 2024-02-15

**Authors:** Xianggang Feng, Yunpeng Wang, Fei Li, Zhifei Hao, Yongfeng Zhang, Yinmin Zhang

**Affiliations:** 1Chemical Engineering College, Inner Mongolia University of Technology, Hohhot 010051, China; feng702520@163.com (X.F.);; 2Key Laboratory of Resource Circulation, Universities of Inner Mongolia Autonomous Region, Hohhot 010051, China; 3ULANQAB Product Quality Measurement Inspection and Testing Centre, Ulanqab 012000, China

**Keywords:** composite materials, styrene butadiene rubber, coal gasification slag, high-value, environmental economy, substitute carbon black

## Abstract

Ultrafine, highly active coal gasification slag (HCGS) was produced via a sustainable, green dry-ball-milling method. Coal gasification fine slag (CGS), a potential environmental pollutant, was used as a new source of rubber filler without pre-treatment, enabling waste utilisation. HCGS was added to styrene-butadiene rubber (ESBR) composites, and the effects of HCGS and the filler content on the mechanical and thermal stabilities of SBR were evaluated. The procedure conforms to important green metrics, requiring no solvent or additional reagent, or solvent-assistance for product collection. HCGS reduced the scorch time (*t*_10_) and curing time (*t*_90_) of the filled ESBR composites relative to those of pure SBR and improved the mechanical parameters. The tensile strength at 50 phr reached 10.91 MPa, and the tear strength at 90 phr reached 64.92 kN/m, corresponding to 9.4- and 3.92-fold increases relative to that of SBR filled with HCGS, respectively. HCGS exerted a reinforcing effect on ESBR, comparable to that of commercial carbon black (CB) N330. HCGS improves the binding between rubber molecules and filler particles and captures the rubber chain, thereby limiting its movement. HCGS is potentially applicable as a CB substitute in the rubber industry, with environmental and economic benefits in the disposal of CGS.

## 1. Introduction

Rubber and plastics are engineering materials used for various parts [1] such as seals, conveyor belts, and automotive industry and hydraulic industry parts. Carbon black is a crucial reinforcing filler owing to its strong interactions with polymers, large surface area, and enhanced surface activity [2]. However, carbon black is produced from non-renewable resources such as oil or natural gas, which consume large amounts of resources and causes high environmental pollution [3]. The search for alternative reinforcing polymers has attracted great attention, not only because of sustainable development and environmental conservation, but also for economic interests [4]. The chemical industry provides important materials and guarantees the energy supply for human, economic, and social activities, inevitably increasing solid-waste discharge. Coal gasification, an advanced and clean coal chemical technology, involves complex reactions in a gasifier. Under high temperatures and turbulent conditions, some fine particles from the crude gas are entrained into the gas-washing and -purification black-water treatment system, generating a large amount of solid waste [5], where coal gasification fine slag (CGS) is produced after filtration. Coal gasification slag comprises mineral particles (MPs) and unburned carbon particles (UCs). The MP glassy microspheres consist of crystalline (silicon, aluminium, calcium, and iron oxides) and vitreous (aluminosilicate) components. Most MPs tend to be smooth and spherical, and some spherical particles are attached to fine debris particles. Inorganic elements are uniformly distributed in the sphere. The molecular formula of UC is determined to be C_178_H_73_O_46_N_1_S_1_, with a molecular weight of 2991. UC has a high degree of aromaticity and abundant oxygen-containing functional groups. UC is loose and porous, which leads to wrapping and coverage with inorganic ash particles [6]. CGS is considered an industrial solid waste that is mainly piled up in slag yards or landfills, thereby wasting carbon resources. Furthermore, the hazardous trace elements in CGFS may pollute soil and groundwater, posing a potential health and environmental risk. Therefore, achieving large-scale, sustainable, efficient utilisation of CGS is urgent.

“Zero emissions” from coal gasification technologies is strongly supported in China. Maximizing coal gasification fine slag (CGS) utilisation and its conversion into high-value-added materials are important for resource utilisation and the harmless disposal of CGS. The utilisation of CGS generally requires copious acid or alkali, or high-temperature and high-pressure treatment. Zheng et al. [7] subjected CGS to low-temperature oxidation at 485 °C for varying residence times under an atmospheric environment, achieving different degrees of activation and yielding activated carbon as an adsorbent. Yang et al. prepared an X-type zeolite/carbon composite via alkaline leaching. Rui et al. [8] synthesised carbon/P-zeolite composites (CPZCs) by using NaOH as an activator for the synthesis of the zeolite, which was thoroughly mixed with CGFS at a certain mass ratio.

Currently, the utilisation rate of coal gasification slag is only 5%. Traditionally, slag is processed into new materials through acidic and alkaline treatments, generating substantial amounts of liquid waste, which is impractical for large-scale industrial use. This study introduces a novel technology for producing polymers from raw coal gasification fine slag, circumventing secondary pollution and affording substantial economic and environmental benefits.

## 2. Material and Methods

### 2.1. Materials

CGS was obtained from an HTL gasifier in the plant at Zhongtianhechuang Erdos Coal Industry Group Co., Ltd. (Erdos, Ordos City, Inner Mongolia Autonomous Region, China). Before the experiment, the ash was dried and dehydrated for 24 h (105 °C). The moisture content of the sample-receiving base was 62.3%, the bulk density of the air-dried base was 382.88 kg/m^3^, the unburned carbon reached 35.3%, and the zeta potential was −21.95. Commercial carbon black (CB; N330) and solid emulsion styrene-butadiene rubber (ESBR, *M_w_* ~400, 000 g) were purchased from Hangzhou Zhongce Rubber Co., Ltd. (Hangzhou, China) and Shandong Gaoshike Industry and Trade Co., Ltd. (Qingdao, China), respectively. The accelerator (*N*-tert-buylbenzothiazole-2-sulfenamide (NS)), zinc oxide (ZnO), stearic acid (SA), and sulphur were obtained from Sinopharm Chemical Reagent Co., Ltd., (Tianjin, China).

### 2.2. Preparation of Ultrafine Coal Gasification Slag (HCGS)

Prior to use, CGS was dried in a high-temperature oven. The dried CGS was put into a polytetrafluoroethylene ball milling tank, after which 1.5 wt% sodium hexametaphosphate solution was added. Ball milling was carried out using the optimised parameters.

### 2.3. Preparation of HCGS/ESBR Composites

The resulting HCGS was directly added to ESBR in an open two-roll mill, similar to a conventional rubber compounding procedure. Other additives (ZnO, SA, NS, and S) were also mixed with SBR and HCGS to form unvulcanised rubber composites. The compounds were vulcanised in a standard mould at 163 °C for the optimal vulcanisation time (determined based on the vulcanisation behaviour). ZnO, SA, the accelerator (NS), and sulphur were sequentially incorporated into the mixture according to the compositions in Table 1.

### 2.4. Characterization

The residual carbon in the slags was measured from the loss on ignition; that is, the relative loss of mass after briefly heating the slags at 815 °C.

The sample (5 g) was weighed, added to 20 mL of ethanol, and then introduced into a laser particle size analyser (Dandong Baite BT800S, Dandong, China) with ultrasound assistance, using ethanol as the dispersant.

The X-ray diffraction (XRD) patterns of the prepared intercalation compounds were obtained using a D8 ADVANCE diffractometer (RIGAKU, Tokyo, Japan) with Cu-Kα radiation (λ = 1.540596 nm; 40 kV and 100 mA) in the range of 5–70° at 10°/min.

Fourier-transform infrared spectra (FT-IR; Magna-IR 750 Nicolet instrument Nigoli Instruments, USA) were recorded at a resolution of 4 cm^−1^ in the range of 4000–400 cm^−1^.

Gold was sprayed (for 30 s) onto a 2 mm slice cut from the tensile fracture interface, which was then treated with a conductive adhesive. Scanning electron microscopy (SEM) images of the composite samples were obtained by scanning 32 times using a Hitachi SU8020 (Hitachi, Japan) cold-field emission scanning electron microscope at an imaging voltage of 30 kV.

The prepared rubber composites were characterised using a (Talos F200X from FEI, Hillsboro, OR, USA) transmission electron microscope at an acceleration voltage of 200 kV. Sections with approximately 50 nm thickness were prepared via ultramicrotomy of the bulk-cured composites.

Raman spectra (Renishaw INVIA microscope laser Raman spectrometer UK) were acquired using a λ = 532 nm laser; the scanning range was 800–2000 cm^−1^.

The zeta potential (−200 mV to +200 mV) and particle size (0.6–12.3 μm) were measured using a Nanoplus3 analyser (Shanghai McMurray Tik Instrument Co., Ltd., Shanghai China) as well as dynamic light scattering (DLS) and electrophoretic light scattering (ELS) at 25 °C.

The processing performance of the rubber composites was tested in accordance with GB/T 9869-2014 [9] using an MDR-2000E non-rotor vulcanising machine (Electronics Wuxi Liyuan Chemical Equipment Co., Ltd., Wuxi, China) at 163 °C, using approximately 5 g of composite material.

The mechanical properties of the rubber composites were tested at a tensile speed of 500 mm/min (WDW-5C electrical tensile tester; Shanghai Hualong Test Instrument Co., Ltd., Shanghai, China) by following GB/T 528-2009 [10] standards.

Cross-linking density test: The cross-linking density of the samples was measured using the swelling method. A tensile sample (thickness: 2 mm, length: 2 cm, width: 1 cm) was weighed (*m*_1_). The sample was immersed in a toluene solution 100 times at a constant temperature for 72 h. The excess toluene on the surface of the sample was removed using filter paper; the sample mass was quickly recorded (*m*_2_). The crosslinking density was calculated using the Flory–Rehner equation [11,12].

The volume fraction (*ν_r_*) of the rubber part in the swollen rubber was determined using Equation (1).
(1)$$v_{r}=\frac{m_{1}/\rho_{r}}{m_{1}/\rho_{r}+m_{2}\rho_{s}}$$

Here, *ρ_r_* is the density of SBR (0.915 g/cm^3^) and *ρ_s_* is the density of toluene (0.866 g/cm^3^).

The rubber crosslinking density *D* was calculated using Equation (2).
(2)$$D=-\frac{\ln\left(1-v_{r}\right)+v_{r}+x{v_{r}}^{2}}{v_{s}(\sqrt[{3}]{v_{r}}-v_{r}/2)}$$

Here, *ν_s_* is the molar volume of toluene (106.87 cm^3^/mol) and *x* is the parameter of the interaction between the ESBR molecules and solvent molecules (*x* = 0.386).

Referring to the Flory–Rehner formula, the effective average molecular weight *M_c_* of two crosslinking points is calculated using Equation (3):(3)$$M_{c}=\frac{\rho_{r}}{D}$$

The swelling rate *Q* is calculated using Equation (4):(4)$$Q=\frac{{(m}_{2}-m_{1})/\rho_{s}}{m_{2}/\rho_{r}}\times 100\%$$

Vulcanisation kinetics: The rate of a chemical reaction can generally be defined as the change in the number of reactants or products per unit of time and space. When other factors are fixed, the mathematical equation that quantitatively describes the influence of the conversion rate of various substances on the reaction rate is called the reaction rate equation. Because the conversion, as a function of time, corresponds to a first-order differential, the vulcanisation kinetics equation can be expressed as Equation (5) [13].
(5)$$\frac{d\alpha }{dt}=K\left(T\right)f(\alpha )$$

Here, *α* is the conversion rate of the vulcanisation reaction. *dα/dt* represents the vulcanisation rate. *T* is the temperature. *K* is the kinetic constant at temperature *T*. *f(α)* is a function of the corresponding phenomenon model. The data obtained from the rotorless vulcaniser test can be used to define α according to Equation (6).
(6)$$\alpha =\frac{Mt-M0}{Mh-M0}$$

*M*0, *M*t, and *Mh* are the torque values at times 0, *t*, and after the vulcanisation reaction, respectively.

As indicated by the vulcanisation curve, the vulcanisation reaction is an autocatalytic process, and the process equation can be expressed by Equation (7).
(7)$$f(\alpha )=\alpha^{m}{\left(1-\alpha \right)}^{n}$$

Both *m* and *n* are the reaction orders. Therefore, this equation can be written as Equation (8).
(8)$$\frac{d\alpha }{dt}=K\left(T\right)\alpha^{m}{\left(1-\alpha \right)}^{n}$$

## 3. Results and Discussion

### 3.1. Physicochemical Characteristics of CGS

#### 3.1.1. Basic Characteristics of CGS

The raw coal fed into the gasifier was a typical blended coal mix of bitumite and anthracite from China. Proximate and ultimate analyses of the CGS samples (Table 2) were performed according to Chinese National Standards GB/T 212–2008 [14], respectively [7]. The ash compositions are listed in Table 3. The gasification slags comprised SiO_2_, Al_2_O_3_, CaO, Fe_2_O_3_, SO_3_, MgO, TiO_2_, and Na_2_O. The high LOI was mainly due to incompletely reacted coal particles, whereas the high contents of Fe_2_O_3_ and CaO were primarily due to the addition of flux agents during the gasification process. The sample had a high ash content (66.88 wt%) and relatively high fixed-carbon content (29.45 wt%).

#### 3.1.2. Morphology Analysis

Figure 1 shows the SEM images of CGS. The micromorphology of CGS is a hybrid of massive porous residual carbon and spherical aluminosilicate slag. The residual carbon is the coal char produced by incompletely gasified syngas and the spherical slag is the fine high-temperature molten coal ash produced by syngas in the gasifier [13]. EDS analysis shows that the spherical particles were enriched with inorganic substances including Si, Al, and Fe. The irregular particles were enriched with unburned carbon. The transmission electron microscopy (TEM) diffraction ring of CGS (Figure 2) shows that the CGS phase is highly disordered with an amorphous structure.

### 3.2. Orthogonal Experiment for Ultra-Fine, Highly Active Coal Gasification Fine Slag

#### 3.2.1. Design of Orthogonal Experiment

The orthogonal design method was used to more accurately and scientifically determine the optimal ball milling parameters by focusing on the material/ball ratio (CGS-Ball milling medium/zirconia ball mass ratio), ball milling time (h), and ball milling rotation using three factors and a five-level orthogonal design *L*_25_ (5^6^). The orthogonal design levels are presented in Table 4.

#### 3.2.2. Mechanical Ball Milling Activation and Refinement of Coal Gasification Fine Slag

Using the orthogonal design table, the design level of each factor in Table 5 was arranged in accordance with the orthogonal design table *L*_25_(5^6^), and 25 matching schemes were created.

#### 3.2.3. Range Analysis of Experimental Results

The orthogonal test indicated that each factor had a different degree of influence on the ball-milling efficiency. The primary and secondary factors were distinguished by calculating and analysing the range R [15,16]. A larger R indicates a more significant influence of the factors on the experimental indices, and vice versa, and can be used to determine the main influencing factor [17].

The calculated data for the milling mechanical activation refinement are shown in Table 6. The ball-milling speed had the greatest influence, followed by the ball-milling time. The ball-to-material ratio had a relatively small influence. During grinding, impact crushing plays a role in the initial stage of crushing the material particles. However, as the particles become finer, crushing and applying force become increasingly unsuitable for ultrafine crushing, particularly for micron-, submicron-, and even nanomaterial particles. The finer the particles, the fewer the cracks and defects, making them harder to grind. Additionally, the exposed, free, electrically charged bonds on the new surface of the particles promoted attraction and agglomeration of the vicinal particles.

#### 3.2.4. Visual Analysis of Orthogonal Test Efficiency Curve for Mechanical Activation and Refinement by Ball Milling

By analysing the results and range of each group of experiments using the orthogonal test, the influence of each factor on the index can be observed. The results are typically intuitively expressed using a factor index diagram. The intuitive trend diagram of the efficiency curve is shown in Figure 3.

The level of each factor is presented on the abscissa, and the average index of each factor is plotted as the ordinate. The best level for each factor was selected to form a combination, corresponding to the optimal reaction conditions.

In the efficiency curve, (1) A1–A5, (2) B1–B5, and (3) C1–C5 represent the average values of each factor level for the medium/ball ratio, rotation speed, and ball milling time, respectively. As shown in diagram (1), when the ball-to-material ratio was A3 (1:4), D90 was the smallest, indicating that the ball-milling efficiency was the highest. When the speed reached B3 (500 rpm), the efficiency was the highest (Figure 2). Similarly, the efficiency was highest when C4 was 4 h (Figure 3). Therefore, the best combination of factors for dry ball-milling is as follows: medium/ball ratio of 1:4, rotation speed of 500 rpm, ball milling time of 4 h.

### 3.3. Effect of Mechanical Refinement by Ball Milling on Particle Size Distribution and Surface Properties of Highly Active Coal Gasification Slag

#### 3.3.1. Changes in Particle Size Distribution and Surface Morphology before and after Ball Milling

The morphology of particles is closely related to numerous particle properties, which is an important aspect of particle characterisation, and directly determines the quality of ultrafine grinding product particles from the perspective of detection. The characterization focused on two aspects: particle size and morphology. Figure 4 shows the change in the particle size distribution of the CGS after mechanical activation by ball milling. After ball milling, the particle size distribution of CGS shifted to the small particle size area, and was centred at 0.5–10 μm, indicating a fractal state. This shows that the CGS is mainly divided into two types of particles: easily broken unburned carbon and breakage-resistant glass beads. However, from the grinding perspective, mixing these two types of materials with different susceptibilities to crushing is beneficial.

As shown in the SEM image (Figure 5), the HCGS particles had no sharp edges or corners after mechanical ball milling, indicative of simultaneous brittle and plastic crushing during mechanical crushing. At the same time, the particles do not easily scratch the rubber particles, which is conducive to reinforcement.

#### 3.3.2. Changes in Surface Properties before and after Ball Milling

Figure 6 shows an infrared scanning image of the coal gasification fine slag after mechanical activation by ball milling. The peak area of the quantitative infrared spectra provides an estimate of the change in the content of each functional group after mechanical crushing. The assignments of the infrared peaks are listed in Table 7.

#### 3.3.3. Changes in Carbon Morphology and Activity before and after Ball Milling

The first-order region in the FT-IR spectrum of carbonaceous materials usually shows peaks at 1620, 1500, 1350, 1200, and 1580 cm^−1^, termed D_2_, D_3_, D_1_, D_4_, and G peaks, respectively [23]. The G peak corresponds to the atomic vibrational mode of the ideal-state graphite lattice (E_2g_ symmetry); D_1_ corresponds to the atomic vibrational model of the disordered graphite structure (graphene layer edges, A_1g_ symmetry); and the D_2_ peak, which is usually accompanied by the D_1_ peak, corresponds to the atomic vibrational mode of the disordered graphite structure (surface graphene layers, E_2g_ symmetry). The D3 peak corresponds to the sp^2^ bonds of amorphous carbon, including organic molecules and functional group fragments, and the D_4_ peak corresponds to the sp^2^–sp^3^ mixed bonds of microcrystalline edges or polyenes formed via C–C and C=C stretching vibrations, including organic molecules and functional group fragments. The D_4_ peak corresponds to mixed sp^2^–sp^3^ bonds at the edges of microcrystals or polyene structures formed via C–C and C=C stretching vibrations. Both the D_3_ and D_4_ peaks arise from reactive sites formed by defects in the carbon structure, which are also known as active sites [23,24,25,26]. The integrated area ratio of the peaks is often used to characterise the structural changes in a sample. The *I_D_*_3+*D*4_*/ I_ALL_* ratio can be used to characterise the active sites of the residual carbon; the higher the ratio, the more active sites are present. However, the *I_D_*_1_*/I_G_* ratio can be used to characterise the degree of structural ordering of the residual carbon; the smaller the *I_D_*_1_*/I_G_* ratio, the higher the degree of structural ordering of the residual carbon [23,27]. The Raman peak fitting (Figure 7) and peak area results are presented in Table 8. *I_D_*_1_*/I_G_* decreased after mechanical activation by ball milling, indicating that the degree of order of HCGS increased after activation, possibly due to the dissociation of the graphite microcrystal layers under mechanical force. *I_D_*_3+*D*4_*/ I_ALL_* increased after mechanical activation, indicating that mechanical ball milling increased the number of active sites in HCGS compared to that in CGS.

#### 3.3.4. Stability of Dispersion System after Mechanical Activation by Ball Milling

Mechanical activation (ball milling) can increase the number of active sites in fine slag powder, reflected in the surface potential (zeta potential) of the resulting ultrafine powder, which is an important index for characterising the stability of powder dispersion systems [28]. Generally, a larger absolute zeta potential indicates greater electrostatic repulsion between the particles and thus better physical stability, that is, better dissolution or dispersion ability [29]. After mechanical activation by ball milling, the absolute zeta potential increased (Table 9), indicating that the electrostatic repulsion of HCGS increased relative to that of CGS, the dispersion was good, and agglomeration was difficult.

### 3.4. Characterization of HCGS-Filled ESBR Composites

#### 3.4.1. Cure Behaviour

During vulcanisation, an essential step in rubber production, the modulus of the rubber rapidly increases [30]. Evaluation of the curing parameters of a rubber compound is necessary to determine the optimum curing time for the fabrication of moulded products, and provides insight into the reinforcement in the moulded compounds.

The curing characteristic curves of the ESBR compounds with different filler fractions at 163 °C are shown in Figure 8. With an increase in the amount of filler, the maximum torque of the composite gradually increased, indicating that the more filler added, the stronger the various network mechanisms inside the rubber.

Typical curing parameters, such as the highest torque (*M_H_*), lowest torque (*M_L_*), and time to achieve 90% of the highest torque (*t*_90_), also known as the optimum curing time, are summarised in Table 10. The M*_H_* value increased with the filler loading at higher filler contents. The impact of the HCGS content on the vulcanization properties of the HCGS/ESBR composite was also assessed. As the filler content increased, *M_H_*, *M_L_*, *ΔM*, and *t*_90_ all exhibited an overall trend of first decreasing and then increasing; *t*_10_ first increased and then decreased. *T*_10_ represents the safety performance of the rubber processing. *T*_90_ represents the rubber processing efficiency. At 30 phr, the processing safety was good and the processing efficiency was high.

#### 3.4.2. Mechanical Properties of ESBR-Based HCGS

The static mechanical parameters of pure ESBR and the composites with different filling fractions of HCGS are listed in Table 11. The 100%, 300%, and 500% tensile stress, and tensile strength first increased and then decreased with an increase in the content of HCGS filler.

The maximum value of 10.91 MPa was reached when the filling amount was 50 phr, which was 84% higher than that of pure ESBR. With an increase in the filling time, the tear strength increased gradually. The maximum tear strength appeared at 64.92 kN/m, which is approximately 29% higher than that of pure ESBR. Owing to the improved interaction between the rubber molecules and HCGS particles, the overall static mechanical properties of the HCGS ESBR composites increased significantly with increasing HCGS content. An increase in the HCGS content can increase the number of HCGS particles per unit volume in the composite system, thus restricting the movement of the rubber chain. To a certain extent, this can be explained by the good mutual attraction between the organic components on the surface of HCGS and ESBR.

#### 3.4.3. Microstructural Properties of HCGS/ESBR Composite

The HCGS/SBR composites were evaluated using SEM (Figure 9) and compared to the non-mechanically activated samples. Numerous rubber chains were enclosed between the HCGS particles, yielding an occluded rubber with restricted rubber chain movement. Additional rigidity was imparted to the rubber molecules, similar to that of the HCGS fillers. TEM images of the composite are presented in Figure 10, revealing that the HCGS particles are arranged in a disorderly manner without any evidence of significant agglomeration below an HCGS particle dose of 50 phr.

### 3.5. Processing Properties and Static Mechanical Properties of HCGS/CB/ESBR Composites with Different Proportions of Carbon Black

#### 3.5.1. Processing Properties of HCGS/CB-Filled ESBR Composites

To improve the static mechanical properties of HCGS/ESBR, ESBR was filled with HCGS and carbon black, and the effects of different compounding ratios on the static mechanical properties of the rubber composites were investigated.

The vulcanisation curves of the ESBR composites with different compounding ratios of HCGS and carbon black are shown in Figure 11. The minimum torques of several composites were not significantly different. The maximum torque of the composites was highest when HCGS/CB = 1:4, indicating the strongest interaction of the filler and ESBR matrix at this ratio.

When HCGS/CB was 1:4, the maximum torque *M_H_* was 1.4, and the difference between the maximum and minimum torques was 1.22 (Table 12). *t*_10_ was 3.30, reaching a maximum, indicating that the processing safety was relatively good, and *t*_90_ was 10.38 min, reaching a minimum, indicating improved processing efficiency. In summary, the comprehensive performance improved after the combination of HCGS and CB, indicating that the combination of HCGS and CB was effective as a filler for reinforcing ESBR.

#### 3.5.2. Mechanical Properties of HCGS/CB-Filled ESBR Composites

The mechanical properties of the ESBR composites prepared with different HCGS/CB ratios are shown in Table 13. At a compounding ratio of 1:4, the tensile and tear mechanical strengths reached a maximum of 12.78 MPa and 90.7 kN/m, respectively. The elongation at break also increased to a maximum of 1301.93. Comprehensive analysis of the mechanical properties showed that HCGS/CB can effectively improve the mechanical properties of ESBR composites.

### 3.6. Crosslinking Density and Average Molecular Weight of HCGS/CB Filled SBR Composites

The crosslinking density and average molecular weight can reveal the number of crosslinks and reflect the interaction of the rubber chains [31]. The crosslinking densities of the ESBR composites with HCGS and CB are listed in Table 14. The filled rubber composites had higher crosslinking densities than pure ESBR, indicating that the enhanced interactions between the HCGS/CB and rubber matrix increased the number of anchor points for the rubber molecules on the surface of HCGS/CB.

The SBR composite filled with HCGS/CB at 1:4 showed the maximum crosslinking density and minimum average molecular weight. Thus, the HCGS/CB composites showed better crosslinking properties, and ultrafine activated coal gasification fine slag may compound carbon black in the filled rubber system, providing more crosslinking sites. This result is consistent with the observed trends in the tensile strength of the SBR composites.

### 3.7. Vulcanization Kinetics of HCGS/CB Filled SBR Composites

Table 15 lists the fitting parameters from Equation (8); k is the rate constant. When HCGS/CB = 1:4, the vulcanization rate was the fastest and k reached the maximum. When the ultra-fine coal gasification fine slag replaced 10 parts of carbon black, the performance of the composite material was not affected and vulcanization was promoted.

## 4. Conclusions

Coal gasification is central to clean coal technology, and the volume of coal gasification slag is expected to increase in the future. Therefore, finding ways to utilise coal gasification slag is currently a top priority. This study is the first to propose the use of mechanically activated coal gasification fine slag as a filler to fill styrene-butadiene rubber-based composites, enabling the utilization of coal-based solid waste and preparation of polymer composites with an excellent performance. The best process route for mechanical activation using a planetary ball mill involved a ball-to-material ratio of 1:4, ball-milling time of 4 h, and ball-milling speed of 500 rpm, at which the ball-milling efficiency was optimal. Quantitative infrared, Raman peak fitting, and zeta potential analyses proved that mechanical activation increases the number of active sites and increases the dispersion of coal gasification fine slag. When HCGS is filled to 50 phr, the static mechanical properties were optimal, and the tensile and tear properties were improved to 10.91 MPa and 64.92 kN/m, respectively. When compounded with carbon black N330, the mechanical properties of the composites improved. When the ratio of HCGS and CB was 1:4, the performance was maximal, and the tensile and tear energy of the composite reached 12.78 MPa and 90.7 kN/m, respectively. The maximum cross-linking density was obtained when the HCGS/CB ratio was 1:4, which was synchronised with the change in the mechanical properties. While this study proposes a method for the large-scale utilisation of coal gasification slag, it is important to focus on future performance degradation mechanisms of rubber products during use.

## Figures and Tables

**Figure 1 polymers-16-00522-f001:**
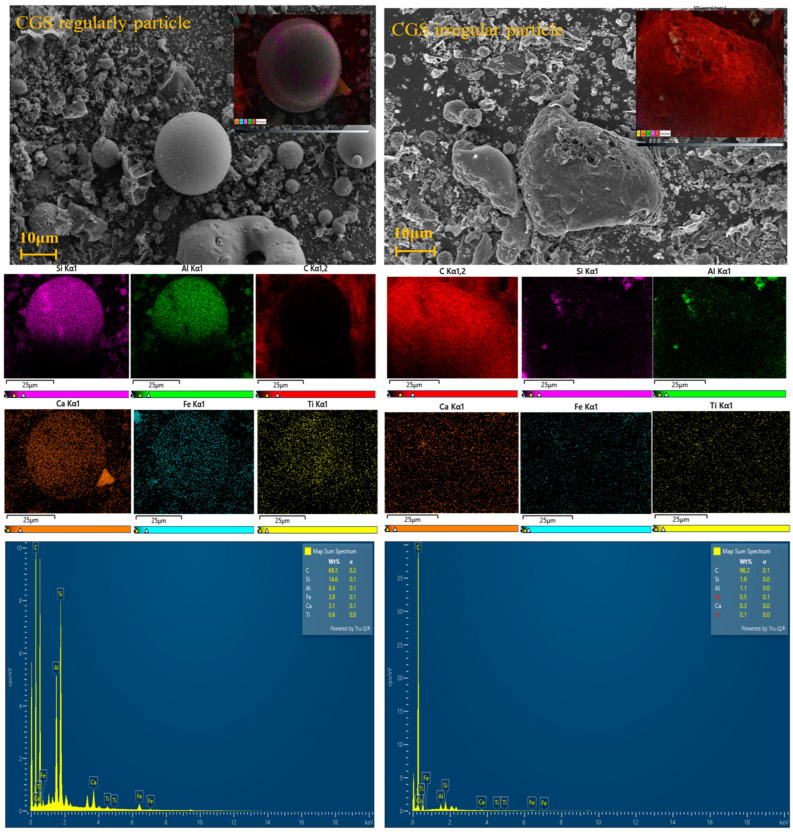
SEM image and EDS energy spectrum of CGS.

**Figure 2 polymers-16-00522-f002:**
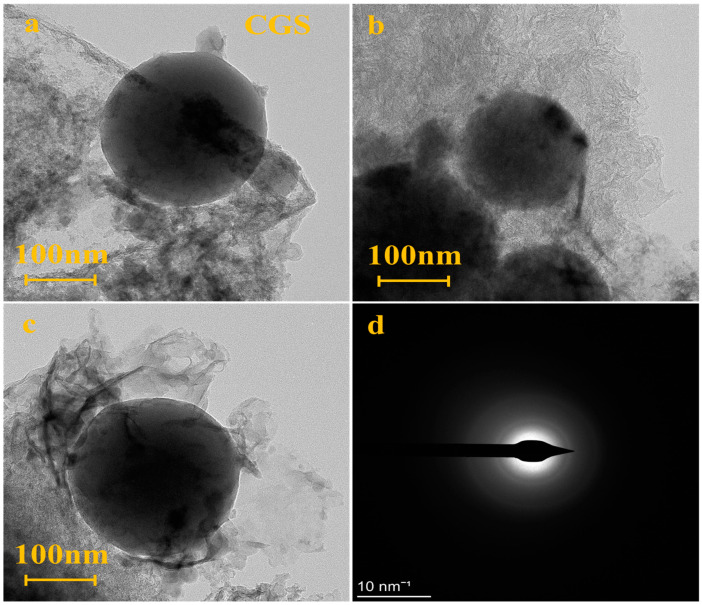
TEM images(**a**–**c**) and diffraction rings(**d**) of CGS.

**Figure 3 polymers-16-00522-f003:**
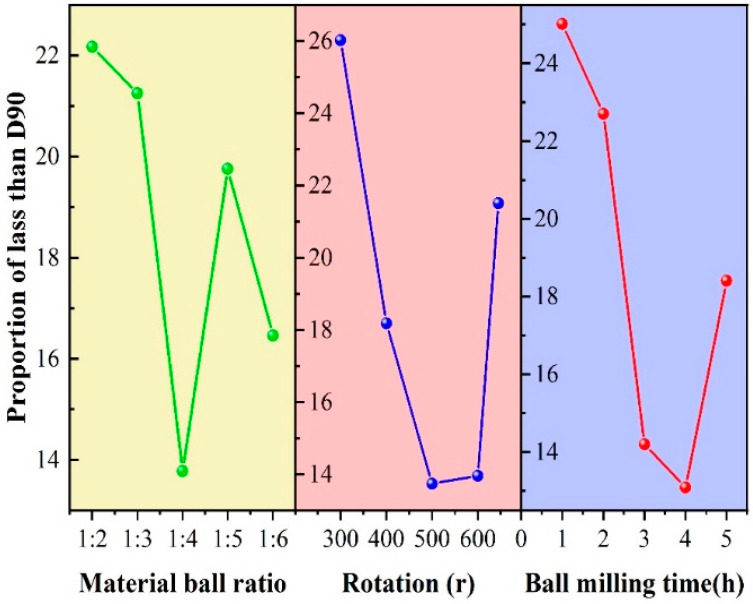
Intuitive analysis of orthogonal experiment efficiency curve.

**Figure 4 polymers-16-00522-f004:**
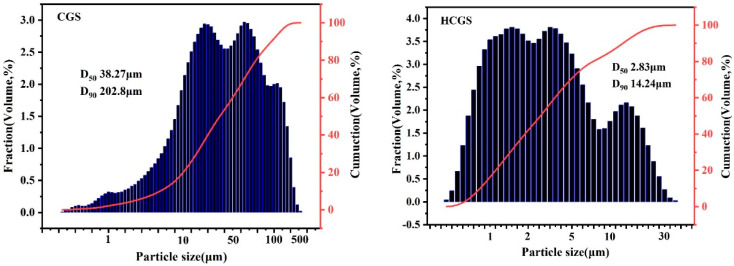
Particle size distribution before and after mechanical activation by ball milling.

**Figure 5 polymers-16-00522-f005:**
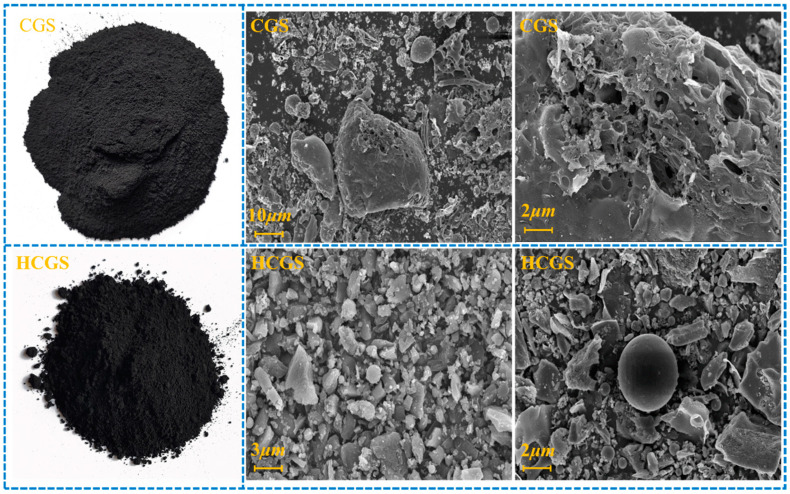
SEM images before and after mechanical activation by ball milling treatment.

**Figure 6 polymers-16-00522-f006:**
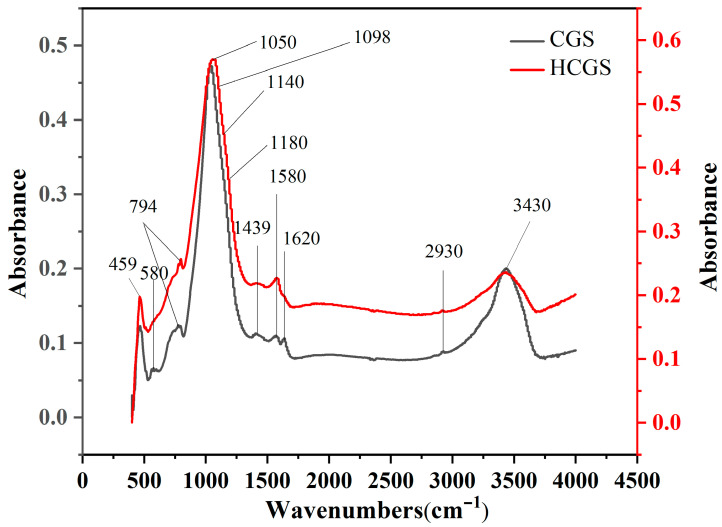
FTIR scanning spectra of CGS and HCGS.

**Figure 7 polymers-16-00522-f007:**
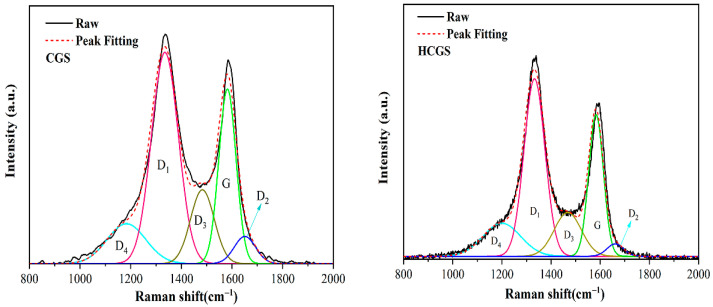
Raman spectra and fitting peaks of CGS and HCGS.

**Figure 8 polymers-16-00522-f008:**
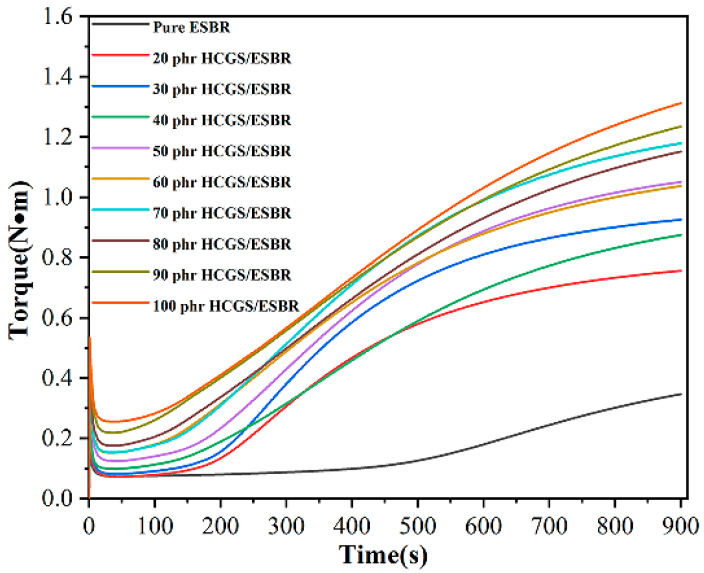
Vulcanization curves of ESBR compounds prepared with different filler contents at 163 °C.

**Figure 9 polymers-16-00522-f009:**
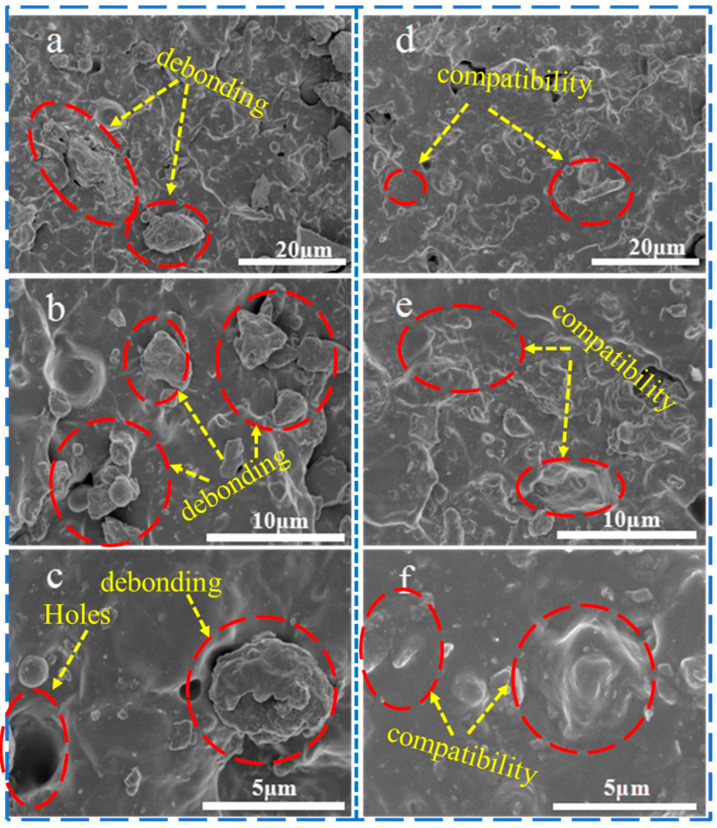
SEM images of HCGS/ESBR composites: (**a**–**c**) CGS/ESBR composite; (**e**–**f**) HCGS/ESBR composite.

**Figure 10 polymers-16-00522-f010:**
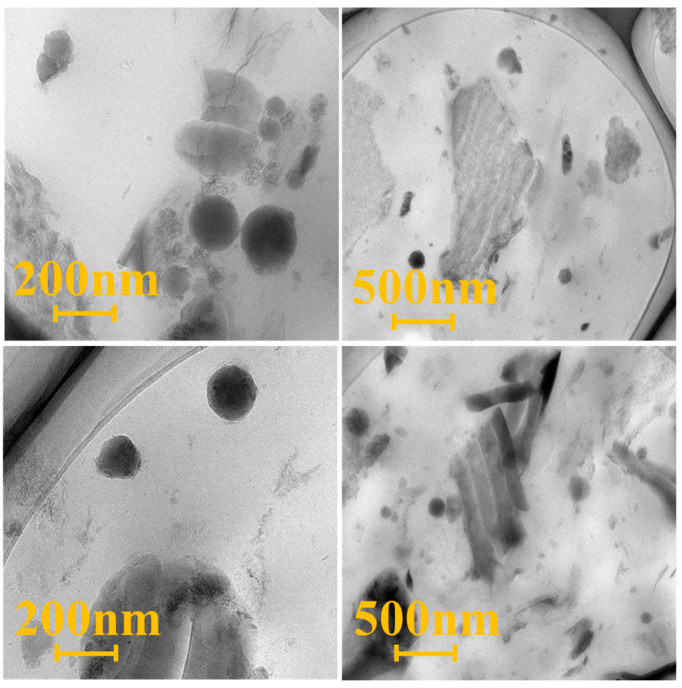
TEM images of HCGS/ESBR composites.

**Figure 11 polymers-16-00522-f011:**
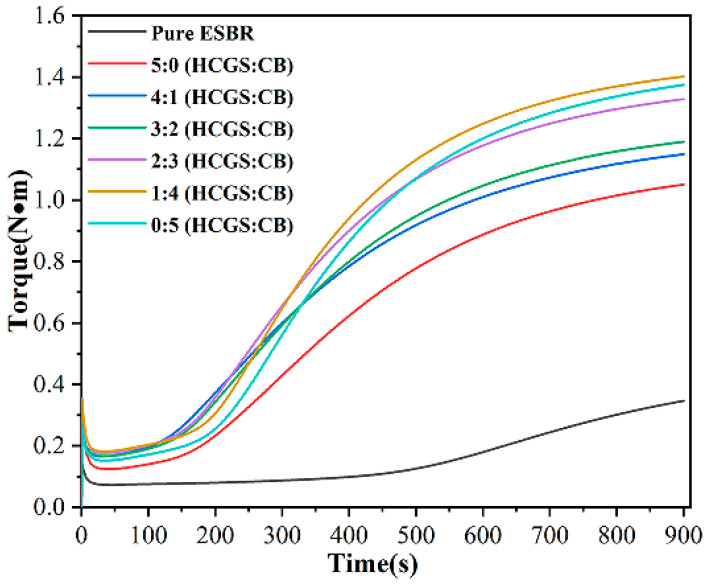
Vulcanization curve of ESBR composite prepared with HCGS and CB at 163 °C.

**Table 1 polymers-16-00522-t001:** Composition of HCGS/ESBR composites.

Component	ESBR	HCGS/ESBR	ZnO	SA	NS	Sulfur
Content/phr	100.00	Variable	3.00	1.00	1.00	1.75

**Table 2 polymers-16-00522-t002:** Proximate analyses of coal gasification fine slag.

Sample (mm)	Mad%	Ad%	Vdaf%	Fcad%
Raw CGFS	0.75	66.88	10.4	29.45

**Table 3 polymers-16-00522-t003:** Chemical compositions of ash from coal gasification fine slag.

Sample	Na_2_O	MgO	Al_2_O_3_	SiO_2_	SO_3_	CaO	TiO_2_	Fe_2_O_3_	LOI_815_/%
CGFS	0.23	1.45	19.6	47.05	4.90	6.77	1.25	11.65	35.3

**Table 4 polymers-16-00522-t004:** Factors and levels of orthogonal experiment.

Factors	Material:Ball Ratio	Ball Milling Time (h)	Rotation (rpm)
Value level	1:2	1	300
	1:3	2	400
	1:4	3	500
	1:5	4	600
	1:6	5	700

**Table 5 polymers-16-00522-t005:** Results of orthogonal array experiment.

Experimental Scheme				Experimental Results		
Serial Number	A Material/Ball Ratio	B (rpm) Rotation	C (h) Ball Milling Time	D10	D50	D90
1	1:2	300	1	2.38	15.25	45.3
2	1:2	400	2	1.53	8.56	22.86
3	1:2	500	3	0.93	4.25	13.34
4	1:2	600	4	0.94	3.46	13.31
5	1:2	700	5	0.89	2.99	16.02
6	1:3	400	1	1.42	9.39	25.43
7	1:3	500	2	0.97	4.48	15.65
8	1:3	600	3	0.89	3.02	13.32
9	1:3	700	4	0.83	2.37	19.28
10	1:3	300	5	1.07	5.60	32.55
11	1:4	500	1	1.01	5.34	15.12
12	1:4	600	2	0.92	3.27	14.49
13	1:4	700	3	0.82	2.27	10.59
14	1:4	300	4	0.91	4.37	15.05
15	1:4	400	5	0.89	2.98	13.64
16	1:5	600	1	0.89	4.04	16.34
17	1:5	700	2	0.92	3.44	38.77
18	1:5	300	3	0.93	4.50	15.47
19	1:5	400	4	0.85	2.51	10.73
20	1:5	500	5	0.84	6.65	17.51
21	1:6	700	1	0.90	4.32	22.88
22	1:6	300	2	1.03	3.99	21.73
23	1:6	400	3	0.87	2.89	18.26
24	1:6	500	4	0.79	2	7.1
25	1:6	600	5	0.76	1.77	12.35

**Table 6 polymers-16-00522-t006:** Range analysis of orthogonal experiment.

D90	K1	22.17	26.02	25.01
	K2	21.25	18.18	22.7
	K3	13.78	13.74	14.20
	K4	19.76	13.96	13.09
	K5	16.46	21.51	18.41
Poor R		8.39	12.28	11.92

**Table 7 polymers-16-00522-t007:** Assignment of absorption peaks in FTIR scanning spectra of CGS and HCGS.

Wavenumber/cm^−1^	Peak Assignment
459	Al-O bending vibration peak [18]
580	Vibration of C–S–C skeleton [18]
794	Si–O–Si [19]
1050	C–O–C [19]
1098	Stretching vibration of Si–O–Al [20]
1140	C–O from alkoxides in RC [21]
1180	Characteristic vibration peak of C–S [18]
1439	C=O bond stretching vibration peak [7]
1580	Characteristic C–C peak of the benzene skeleton [18]
1620	C=C bond stretching vibration peak [20]
2930	Vibration of C–H [22]
3430	OH bonds [22]

**Table 8 polymers-16-00522-t008:** Area ratio of Raman bands.

Peak Type	Area (CGS)	Area (HCGS)
D_4_	38,947.6711	32,370.39397
D_1_	328,489.60646	152,371.04697
D_3_	47,477.81474	29,243.27679
G	161,914.21121	86,728.18662
D_2_	2192.3369	6971.39049
*I_D_* _1*/G*_	2.029	1.757
*I_D_* _3*+D*4*/*_ *I_ALL_*	0.1493	0.2003

**Table 9 polymers-16-00522-t009:** Zeta potential before and after mechanical activation (ball milling).

Sample	Zeta Potential
Raw CGS	−21.95
HCGS	−26.05

**Table 10 polymers-16-00522-t010:** Vulcanization parameters of ESBR compounds prepared with different filler contents at 163 °C.

Vulcanization Index (phr)	*M_L_* (dN·m)	*M_H_* (dN·m)	*M_H_*-*M_L_* (dN·m)	*t*_10_ (min)	*t*_90_ (min)
Pure ESBR	0.07	0.35	0.28	6.49	13.56
20	0.07	0.76	0.69	3.26	11.1
30	0.08	0.93	0.85	3.27	10.52
40	0.10	0.87	0.77	3.09	12.19
50	0.12	1.05	0.93	3.1	11.32
60	0.15	1.04	0.89	2.36	11.39
70	0.15	1.18	1.03	2.5	11.43
80	0.18	1.15	0.97	2.38	12.18
90	0.22	1.23	1.03	2.25	12.28
100	0.26	1.31	1.05	2.46	12.43

**Table 11 polymers-16-00522-t011:** Mechanical parameters of ESBR composites with different HCGS filling fractions.

Different Fillers	Tensile Strength/MPa	Tear Strength/(KN/m)	Modulus/MPa			Elongation at Break/%
			100%	300%	500%	
Pure ESBR	1.16 ± 0.133	16.57 ± 2.440	0.43 ± 0.016	0.56 ± 0.005	0.60 ± 0.008	786.72 ± 139.7
20 phr	6.09 ± 1.471	34 ± 3.347	0.85 ± 0.019	1.35 ± 0.013	2.06 ± 0.064	915.54 ± 82.35
30 phr	8 ± 0.578	42.25 ± 8.313	0.94 ± 0.027	1.56 ± 0.061	2.49 ± 0.179	985.12 ± 54.6
40 phr	10.48 ± 0.973	47.41 ± 5.897	0.98 ± 0.041	1.51 ± 0.097	2.39 ± 0.27	1337.18 ± 148.4
50 phr	10.91 ± 1.877	49.01 ± 6.341	1.17 ± 0.071	2.05 ± 0.185	3.48 ± 0.483	1023.60 ± 144.3
60 phr	9.96 ± 0.372	57.04 ± 3.307	1.22 ± 0.03	1.99 ± 0.069	3.28 ± 0.187	1160.44 ± 63.05
70 phr	9.97 ± 0.25	58.21 ± 2.981	1.4 ± 0.078	2.48 ± 0.251	4.28 ± 0.593	991.52 ± 132.7
80 phr	6.96 ± 0.206	62.34 ± 2.891	1.45 ± 0.079	2.33 ± 0.163	3.61 ± 0.235	999.22 ± 65.3
90 phr	7.87 ± 0.276	64.92 ± 2.896	1.71 ± 0.081	2.86 ± 0.11	4.54 ± 0.201	907.16 ± 17.24
100 phr	6.53 ± 0.392	63.02 ± 2.272	1.83 ± 0.063	3.03 ± 0.147	4.48 ± 0.305	818.10 ± 66.21

**Table 12 polymers-16-00522-t012:** Vulcanization parameters of ESBR composite with HCGS and CB prepared at 163 °C.

Vulcanization Index	*M_L_* (dN·m)	*M_H_* (dN·m)	*M_H_-M_L_* (dN·m)	*t*_10_ (min)	*t*_90_ (min)
pure SBR	0.07	0.35	0.28	6.49	13.56
HCGS:CB 5:0	0.18	1.05	0.87	2.54	12
HCGS:CB 4:1	0.17	1.15	0.98	2.32	11.01
HCGS:CB 3:2	0.17	1.19	1.02	2.49	10.57
HCGS:CB 2:3	0.18	1.33	1.15	2.55	10.45
HCGS:CB 1:4	0.18	1.40	1.22	3.30	10.38
HCGS:CB 0:5	0.15	1.37	1.22	3.30	10.58

**Table 13 polymers-16-00522-t013:** Mechanical properties of ESBR composites prepared with HCGS and CB.

Different Fillers	Tensile Strength/MPa	Tear Strength/(KN/m)	Modulus/MPa			Elongation at Break/%
			100%	300%	500%	
Pure SBR	1.16 ± 0.133	16.57 ± 2.440	0.43 ± 0.016	0.56 ± 0.005	0.60 ± 0.008	782.72 ± 139.7
HCGS:CB 5:0	10.91 ± 1.877	49.01 ± 6.341	1.17 ± 0.071	2.05 ± 0.185	3.48 ± 0.483	1023.60 ± 144.3
HCGS:CB 4:1	10.89 ± 0.297	64.67 ± 12.82	0.96 ± 0.205	2.11 ± 0.276	3.95 ± 0.266	1259.35 ± 70.65
HCGS:CB 3:2	11.12 ± 0.77	69.74 ± 10.56	0.96 ± 0.097	1.87 ± 0.109	3.4 ± 0.275	1286.32 ± 84.89
HCGS:CB 2:3	11.89 ± 0.442	82.91 ± 6.299	1.37 ± 0.121	2.58 ± 0.135	4.44 ± 0.159	1285.45 ± 44.82
HCGS:CB 1:4	12.78 ± 0.42	90.70 ± 9.996	1.14 ± 0.012	2.42 ± 0.111	4.36 ± 0.259	1301.93 ± 88.55
HCGS:CB 0:5	12.52 ± 0.007	89.55 ± 6.583	1.15 ± 0.007	2.1 ± 0.106	3.79 ± 0.247	1289.23 ± 39.28

**Table 14 polymers-16-00522-t014:** Crosslinking density of ESBR composites with HCGS and CB.

Type of Modifier	Crosslinking Density (×10^−4^ mol/cm^3^)	Swelling Ratio (%)	Available Average Molecular Weight (×10^−4^ g/mol)
Pure SBR	0.3330	96.4582	0.02748
HCGS:CB 5:0	2.1678	75.6507	0.4221
HCGS:CB 4:1	2.2885	79.1478	0.3998
HCGS:CB 3:2	2.3506	78.7522	0.3893
HCGS:CB 2:3	2.7835	76.6002	0.3287
HCGS:CB 1:4	2.9364	75.2460	0.3116
HCGS:CB 0:5	2.8975	76.3226	0.3263

**Table 15 polymers-16-00522-t015:** Kinetic parameters obtained by fitting the experimental curing data to Equation (8).

ParpaleCB (phr)	*K*	*m*	*n*	*R^2^*
0	0.00384	0.68414	0.71796	0.85805
10	0.00434	0.50127	0.82108	0.96624
20	0.00565	0.71001	0.92256	0.96605
30	0.00702	0.83906	1.01456	0.97042
40	0.01197	1.20423	1.25499	0.94793
50	0.01118	1.15232	1.22475	0.93476

## Data Availability

The data presented in this study are available on request from the corresponding author.

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
