# Peer review of "Performance of Full-Component Coal Gasification Fine Slag: High-Value Utilization as Reinforcing Material in Styrene-Butadiene Rubber (ESBR) for Replacing Carbon Black"

_polymers, 2024, doi:10.3390/polym16040522_

Round 1

Reviewer 1 Report

Comments and Suggestions for Authors

The manuscript needs extensive revision to be accepted for publication in any journal. Some comments are given below:

Abstract

1 An ultrafine, unwanted usage of capital A. Such usage is there throughout the manuscript. 

2. What is ESBR? What is E here. 

3. tensible?

4. Mpa should be MPa. This is applicable in many places in the manuscript. 

5. What you mean by capturing rubber chains?

6. Abstract needs to be modified. Improvement in mechanical properties is repeated. 

Introduction

7. Line 38: Materials and methods 2?

8. What is MP?

9. What is CGFS?

10. No real literature review is provided about CGS, its utilization in polymers as fillers etc. 

11. HTL?

12. Line 109-110: Which machine is used?

13. line 111: What type of measurement is done here?

14. line 114: parameters for SEM measurement not given. 

15. line 117: 200 KV, should be kV
16. line 147: conversion rate and conversion rate?

17. line 169: CGFS should be CGS

18. Figure 1. Which technique is used for this measurement? Not mentioned anywhere.

19. line 182: Fig shows the SEM Fig.2. What do you mean?

20. Line 223: Tible 5?

21. Line 230: free electricity bond. What does it mean?

22. Figure 1 and Figure 5 first part are same. Delete figure 1. Combine the texts related to Figure 1 with section 3.3.1

23. Table 6: if C=O is present a sharp peak around 1700 cm-1 should be seen. I could not see any such peak in FTIR curve. Why so?

24. Zeta potential measurement is not described anywhere. 

25. line 339: Table number is missing

26. line 341: As filler content increased, repeated. 

27. Figure 10 and Table 10 are one and the same. Delete Figure 10. 

28. Delete figure 14.

29. The properties with CB is greater than HGS as per table 12. The effectiveness of HCGS is not so good compared to CB. 

30. It is very difficult to understand the mechanism of reinforcement between HCGS and rubber as written now. I don't believe there is any interaction between the filler and the rubber molecules. 

Comments on the Quality of English Language

Quality of English is very bad. It was very difficult for me to understand what the authors want to convey. 

Reviewer 2 Report

Comments and Suggestions for Authors

Zhang et al propose the use of fine slag as filler in ESBR matrix as an alternative to the traditional filler such as CB (N330). The study is interesting and nicely formulated. Various experimental tasks are carried out such as rheometric curves, crosslink density and mechanical properties. The results are nicely presented and nicely established with key take away from this interesting study in conclusion section. However, some revisions are required before possible publication. Some points are –

[1] Why author abbreviate ESBR from styrene butadiene rubber? Is it Emulsion-SBR? If yes, please write it in the first abbreviation in abstract?

[2] The English of the paper needs substantial revision from the native speaker. Many sentences are incomplete or complex with no meaning or grammatically incorrect, So, please make the English editing. For example – In introduction #line 38, “Material and methods2”- what does it mean here? Moreover, All the references must be edited and cited correctly according to journal format.

[3] In last paragraph of introduction, please highlight the novelty of present work such as what is the main theme of the work? why this work is important and what are its advancement from existing literature must be stated clearly.

[4] In section 2.3, the “formulation” of the composites should be reported in table form. Moreover, why authors choose these quantities for the compositions such as do they are optimized must be reported?

[5] In section 2.4, how SEM sample was performed? How coating of sample was performed? How mechanical properties were performed? What standards were followed to test mechanical properties? How much was the load cell and pre-load? What was the strain rate and sample dimensions etc are missing?

[6] Figure 2 need to be more clear. For example, the details of elemental compositions and their respective mapping is not visible at all. Please zoom them and make them visible.

[7] In Figure 9, the font size of the compositions is not visible. In section 3.4.2 and 3.5.2., please provide the stress-strain curves for all compositions. In Figure 10, the y-axis should be “Tensile strength”

[8] The conclusion should be further improved by adding current trends, challenges and future prospects of the work.

Good Luck with the revisions!

Comments on the Quality of English Language

Extensive editing of English language required

Reviewer 3 Report

Comments and Suggestions for Authors

Novelty of the paper is moderate. Adding slag to numerous materials is quite commo, however metodology is correct. There are numerous small errors., for example:

1. Raman intensity has values but not units. I assume that they are counts, but exact values do not give useful information. I would remove values for Y-axis.

2. Table 6 - numerous editorial errors [e.g. spaces, references]. Similair, numerous errors in whole document.

3. Numerous errors with indexes, e.g.  1200 cm−1.

Extensive editing of whole paper is  required.

Comments on the Quality of English Language

Novelty of the paper is moderate. Adding slag to numerous materials is quite commo, however metodology is correct. There are numerous small errors., for example:

1. Raman intensity has values but not units. I assume that they are counts, but exact values do not give useful information. I would remove values for Y-axis.

2. Table 6 - numerous editorial errors [e.g. spaces, references]. Similair, numerous errors in whole document.

3. Numerous errors with indexes, e.g.  1200 cm−1.

Extensive editing of whole paper is  required.

Round 2

Reviewer 1 Report

Comments and Suggestions for Authors

The comments were addressed diligently. 

Please check Table 11 units again.

Author Response

I appreciate the valuable comments from the reviewer. I apologize for this mistake Here is my answer to the question:

I have amended the Mpa in Table 11 to MPa.

Reviewer 2 Report

Comments and Suggestions for Authors

Minor revisions requested as -

[1] In Table 1, please expand ESBR, HCGS/ESBR, ZnO, SA, and NS below the table.

Author Response

I appreciate the valuable comments from the reviewer. Here is my answer to the question:

In Table 1, I have thickened the ESBR, HCGS / ESBR, ZnO, SA, and NS fonts.